# Influence of Work-Related Safety and Health Guidelines on Knowledge and Prevalence of Occupational Back Pain among Rehabilitation Nurses in Saudi Arabia: A 6-Month Follow-Up Study

**DOI:** 10.3390/ijerph18168711

**Published:** 2021-08-18

**Authors:** Ahmad H. Alghadir, Hani Al-Abbad, Syamala Buragadda, Amir Iqbal

**Affiliations:** 1Rehabilitation Research Chair, College of Applied Medical Sciences, King Saud University, Riyadh 11433, Saudi Arabia; alghadir@ksu.edu.sa (A.H.A.); syamala3110@yahoo.co.in (S.B.); 2Physical Therapist, King Fahad Medical City, Riyadh 12231, Saudi Arabia; haniala733@gmail.com

**Keywords:** occupational safety and health, risk assessment, occupational disorder, knowledge, low back pain, rehabilitation nurses, patient care

## Abstract

Background: Nurses are frequently involved in different types of patient handling activities in different departments of the hospitals. Mishandling the patients causes accumulative stress on their spine that results in occupational back pain (OBP), substantial morbidity, and incurred cost. Objectives: This study aimed to observe the influence of work-related safety and health guidelines on knowledge and prevalence of occupational back pain among rehabilitation nurses in Saudi Arabia. Methodology: This cohort study was conducted with the inclusion of a total of 116-registered rehabilitation nurses (97-female, 19-male, mean age = 39.6-years) from different regions of Saudi Arabia. After the invitation, these nurses attended an ergonomic workshop focusing on work-related safety and patient handling guidelines, risk assessment, and control of OBP. A self-administered questionnaire was used to assess the knowledge, risk, and prevalence of OBP at baseline and 6-months follow-up. Results: The perceived knowledge score significantly improved (95% CI; t = 4.691; *p* < 0.001; Cohen’s *d* = 0.72) at 6-month follow-up (mean ± SD = 81.6 ± 18.2) from its baseline score (mean ± SD = 68.2 ± 19.2). Likewise, the prevalence score of OBP markedly reduced from 71.5% (baseline) to 65.0% (6-month follow-up). Conclusion: The level of knowledge highly improved and the prevalence of OBP markedly reduced within a span of 6-month among rehabilitation nurses in Saudi Arabia after attending an ergonomic workshop. Importantly, the nurses learned and geared up themselves for practicing the safe patient handling guidelines to avoid occupational back pain in the future. Therefore, rehabilitation nurses should update their knowledge and awareness about occupational safety and health guidelines, risk assessments, and control of OBP at a regular interval for increasing the knowledge and reducing the prevalence of OBP among them.

## 1. Introduction

Occupational back problems among nurses is an area of interest for many authors in different countries as they account for substantial morbidity and cost [1]. Nurses are frequently involved in different patients handling activities that require either prolonged sustained postures or repetitive movement that have an accumulative stress on the spine [2,3]. Many risk factors have been identified as contributing to this problem including increased physical work demands, nurses’ skills in patient handling, poor ergonomics in patient care, unavailability of assistive devices, psychosocial factors, and work organizational factors [4,5]. Interventions to help prevent or reduce this problem have shown controversial results. Among these interventions were patients handling education and training which is the most common method, ergonomics intervention, lifting teams, stress management, exercises, and provision of assistive devices [6,7].

The literature is abundant with studies that used single-factor interventions that showed limited impact on outcomes [8,9,10,11]. However, studies utilizing multidimensional strategies specifically based on risk assessment and control are more likely to be effective [12,13]. The aim of the risk assessment process is to identify the potential risks involved in patient handling and subsequently control them. The risk control process requires taking all available steps to eliminate hazards, if it is not possible to isolate them, they must be minimized and closely monitored for their effectiveness [14,15]. Four key risk factors need to be assessed [15]. First, the load refers to “patient characteristics” that can affect the handling risk. Second, the individual refers to the capabilities of the caregiver that can influence their capacity to carry out the job safely. Third, the task refers to the nature of the task as different tasks with different requirements, each needing proper assessment and a unique approach. Fourth, the working environment that impacts how the task is performed. The process takes place at the levels of the workplace and in relation to the handling of each patient [15].

The influence of nurses’ awareness and knowledge about safe patient handling practice and compliance to standard guidelines is scarce in the literature and advised by many authors to design such an educational program scheduled with their work as to avoid the work-related injuries among them [16,17]. To our knowledge, there are not enough studies that have approached multidirectional strategies, including organized courses with hands-on-workshop to improve the knowledge and awareness about risk assessments and their control in reducing the prevalence of work-related low back pain among nurses. This study fulfills that scarcity by estimating the effectiveness of a well-organized ergonomics workshop focused on safe patient handling guidelines and measures to reduce the prevalence of occupational back pain among rehabilitation nurses. Furthermore, this study points out the importance of adhering to a particular safe patient’s handling guidelines and measures aiming to be free from occupational back pain among rehabilitation nurses.

### Objectives

The objective of this study was to educate and increase the level of knowledge and awareness about the risk assessments and control of OBP among rehabilitation nurses in Saudi Arabia through following the safe patient handling guidelines. In addition, it also aimed to evaluate the impact of the ergonomics workshop on the level of knowledge and the prevalence of OBP among them. Two research questions/hypotheses directed this study as follows:

Did the level of acquired and perceived knowledge about the risk assessments and control of OBP among rehabilitation nurses in Saudi Arabia increase after attending the ergonomics workshop?

Did the improved acquired and perceived knowledge (about the risk assessment and control of OBP) after attending the ergonomics workshop play an important role in reducing the prevalence of OBP among rehabilitation nurses in Saudi Arabia?

## 2. Materials and Methods

### 2.1. Study Design

A cohort study design with six-months follow-up was used in this study.

### 2.2. Ethical Consideration

This study was fully complied with the ethical standard for human research and approved by the Ethics Sub-Committee at King Saud University (file ID: RRC-2017-003; dated: 23-02-2017) and also complied with the Helsinki Declaration as revised in 2013.

### 2.3. Setting

An invitation to attend the ergonomics workshop titled “workshop on patient’s handling and occupational back pain among rehabilitation nurses” was sent via email to different hospitals in Riyadh city, Saudi Arabia.

### 2.4. Participants

Practicing nurses handling patients and head and in-charge nurses were the targeted participants in this study. Inclusion criteria were included nurses working in rehabilitation, having the age more than thirty years, and continuous practicing experience of more than two years. Nurses who had morbid obesity and health-related problems prohibiting the provision of patient care were excluded from the study. Out of the 156 nurses who registered for the workshop, 116 were screened and recruited for the study based on the inclusion and exclusion criteria.

### 2.5. Procedure and Measurements

A one-day ergonomics workshop was conducted at a specified center of our university. It was organized and accredited for a 7 h continuing medical education (CME) by our university. The course delivered in the ergonomic workshop was divided into theoretical and practical sessions. The theoretical topics included anatomy and risk of injury, biomechanics, and the concept of patient handling based on the “New Zealand Patient Handling Guidelines-the LITEN UP Approach 2003”, (Wellington, New Zealand, 2003) evidence-based patient handling, controversial techniques, and hazardous tasks, walking aids and patient handling assistive devices, back care, and exercises [15,17]. The practical session covered the use of walking aids and assistive devices, techniques of safe patient handling including moving the patient in bed, bed positioning, sitting to the edge of the bed, standing, and sitting, bed to wheelchair transfer, transferring a patient on lying surface, assisting a fallen patient. The concept of risk assessment and control was the theme of instruction during the entire workshop. The workshop was delivered by four instructors experienced in back care and patient handling. The participants were provided with a 30 pages’ manual covering all aspects of the workshop.

### 2.6. Outcome Measures

A “self-administered questionnaire” of two pages was given to the participants on the day of the workshop as a baseline measure of their knowledge, risk, and exposure to occupational back pain. The initial version of the questionnaire was developed based on a literature review. The questionnaire was then reviewed by a panel of experts with more than 10 years of experience. Modifications were made based on the recommendation of the panel. A convenience sample consisting of 20 non-participating nurses was asked to fill out the final version of the questionnaire on two occasions, two days apart. Test-retest reliability was assessed using intra-class correlation coefficient, which ranged between 0.85–0.9, indicating high reliability of the questionnaire.

A cover letter stated the objective of the study and assured the participants that the data obtained are confidential. Each questionnaire was coded with a number that corresponded to a master list of names. The questionnaire was composed of four main sections with primarily “close ended questions”. The first section was designed to obtain demographic information such as gender, age, height, weight, educational level, years of professional experience, and type of working area. The second section aimed to evaluate the perceived level of knowledge and awareness of the participants about different parameters derived from standard guidelines of safe patient handling and previous training on patient handling. The third section inquired about the participant’s physical exposure including the number of patients handled on a daily basis, the percentage of time devoted for patients moving and transfer, and the handling tasks practiced during work. The Standardized Nordic Questionnaire was used in the fourth section to assess the amount of back injury through information on the number of days with back problems during the past year [18].

A 6-month follow-up questionnaire was presented to the participants through their e-mail contacts. The baseline questionnaire items were preserved with two additional items in the second section asking about the perceived implementation of knowledge gained from the workshop and reasons for not implementing the knowledge gained in the workshop into practice. The third section was replaced with a quiz of 10 true/false questions to examine the knowledge gained at the workshop. The Nordic questionnaire was modified to ask about back injuries during the past 6-months.

### 2.7. Statistical Analysis

Data analysis for all the variables was done using the statistical software SPSS (IBM SPSS Statistics for Windows, v. 21, Armonk, NY, USA, IBM Corp). Mean differences from baseline to 6-months follow-up and descriptive statistics were calculated by applying a paired t-test. Further, Cohen’s *d* [19] test was used to see the effect size of the intervention (the ergonomic workshop) on the perceived knowledge among the participants. Percentage change and composite means of the test scores were used to evaluate the prevalence of OBP, implementation of perceived knowledge, and acquired knowledge for all the participants. The level of significance (α) was set at 0.05 for all the statistical analyses.

## 3. Results

### 3.1. Demographic Characteristics

The analysis was conducted on the subset sample of participants that responded to the second questionnaire. A total of 116-participants attended an ergonomics workshop to receive an educational intervention (the ergonomic workshop) but only 84-participants returned the questionnaire with a response rate of 72.4% at 6-months follow-up via e-mail. The mean age of the participant nurses was 39.6 ± 8.60 years, with a BMI of 26 ± 4.70 kg/m^2^. Their average clinical work experience was 5 ± 1.50 years. In addition, data for professional characteristics including educational level, area/department of practice, and year of professional experiences are presented below (see Table 1).

The outcome measures of this study were perceived and acquired knowledge and the prevalence of OBP, and the independent variables were educational intervention in the ergonomic workshop and physical exposure parameters.

### 3.2. Perceived Knowledge

As shown in Table 2, the score of perceived knowledge among nurses was significantly improved (13.4 ± 9.5; *p* < 0.05) from its baseline score (68.2 ± 19.2) to 6-months of the follow-up score (81.6 ± 18.2). Furthermore, an item-wise and overall comparison of perceived knowledge can be seen between baseline and 6-month follow-up (see Figure 1 and Figure 2).

In addition, a Cohen’s *d* test applied indicating a large effect-size (Cohen’s *d* = 0.72) of an ergonomics workshop on the perceived knowledge among rehabilitation nurses (see Table 3).

**Table 3 ijerph-18-08711-t003:** Comparison of the effects between pre- and post-ergonomic workshop on the perceived knowledge (Cohen’s *d* and paired *t*-test).

Variables	Mean ± SD	Paired *t*-Test		Cohen’s *d*
		t-Value	*p*-Value	
**Pre-workshop**	68.2 ± 19.2	4.691	0.001 **	95% CI [3.18, −4.21] *d* = 0.72
**Post-workshop**	81.6 ± 18.2			
**(M2-M1)/SD pooled**	13.4/18.71			

** extremely significant if *p* < 0.001; small effect if *d* = 0.20; medium effect if *d* = 0.40; large effect if *d* = 0.60 [19].

### 3.3. Acquired Knowledge

Test yourself. Ten questions were asked to assess the extent of knowledge gained from the course delivered in the workshop and the composite score mean obtained was 7.67 ± 1.10. Their test score indicated that the level of knowledge acquired from the course delivered in the workshop was above the average. Prior to distributing the questionnaire to the participants, we instructed them to fill the answers at their own acquired knowledge from the course without referring back to the manual or other nursing professionals.

### 3.4. Perceived Implementation of Knowledge

The majority of participants (98.0%) declared “my manual handling knowledge improved after taking the ergonomics workshop”. Around 90.0% of participants agreed that “I applied gained knowledge into my daily work” and that probably affected the percentage reduction in the prevalence of their occupational back pain among them.

### 3.5. Prevalence of OBP

The prevalence of OBP symptoms reduced with a difference of 6.5% (65% at 6-months follow-up) from its baseline scores (71.5%). This confirms that the nurses applied the gained (13.4 ± 9.5; *p* < 0.05; Cohen’s *d* = 0.72) perceived knowledge (after the ergonomic workshop) into their clinical practices while handling the patients. Moreover, this also confirms the impact/role of the ergonomics workshop (an educational intervention) on reducing the prevalence of OBP among rehabilitation nurses in Saudi Arabia.

## 4. Discussion

The present study was designed to investigate the impact of a workshop-based educational program on the knowledge and awareness about potential risk factors on the prevalence of occupational back pains. The prevalence of low back pain in Saudi Arabia ranges from 53.2% to 79.2% and has multifactorial risk factors such as vitamin deficiency, obesity, sprains, stretching, and bending activities [20]. In addition, the number of patients, number of working hours in patient handling, patients care with poor ergonomics, and lack of adoption of “no lift policies” by health organizations were marked as major risk factors for back pain related to occupation [21]. Similarly, Mitchell et al. (2008) reported that noticeable health problems were identified “higher” among the nurses working longer than “20 h per week” in the hospitals, and about 60% of them were taken treatment, medicine, or reduction in activities [22]. It is widely known that physical exposure is the strongest risk factor of back pain among nurses and other health care workers [16]. However, means of prevention remain controversial. Nurses’ injuries develop with the use of improper techniques in lifting and handling, which cause unnecessary stress and load on the spine and results in LBP; once an injury takes place, re-injury is inevitable [23,24]. Likewise, many studies supported this view that direct patient handling during the provision of care such as shifting (bed sheath changing, side rotation, up and down), lifting (heavy patients), mobility (assisted walking), transfer (bed to wheelchair, wheelchair to vehicle, etc.) and toilet care activities constitute a major risk for occupational low back pain among nurses in the hospitals [7,12,25].

The literature is abundant with evidence on back injuries among nurses [26,27,28]. Such injuries are associated with sick leaves [29]. LBP has been linked to worker’s compensation claims and disability insurance in Western countries [30]. There is strong evidence from four high-quality studies and eight moderate studies that training intervention has no impact on working practice and injury rate. However, other studies showed that training interventions have mixed (positive and negative) short-term results [31]. An intervention based on a risk assessment program is most likely to be beneficial in reducing risk factors during patient handling [32]. We believe that there is a strong need to address new interventional preventive strategies emphasizing on risk assessment and control principle that looks at this problem from a multidimensional view.

Our findings confirm the roles of knowledge and awareness and the importance of education in reducing the risk of back pain among nurses [33,34]. Moreover, this is in line with previous studies [12,25]. The decline in self-reported back pain from 71.5% to 65% observed in this study 6-months after attending a workshop on safe handling techniques reflects the need for ongoing training. The value of introducing a patient handling policy and compliance with guidelines at work remains to be explored. We support the current moderate evidence that supports utilizing multidimensional strategies specifically based on risk assessment and control strategies that could potentially produce favorable results [6,7,35].

The risk of musculoskeletal injuries is mostly associated with dependent patient care and is usually secondary to manual patient handling. Knowledge about how and when to use assistive devices is necessary to avoid back injuries as high forces are required to transfer patients [36]. In contrast, ergonomic training proved ineffective in the prevention of back injuries with manual patient lifting [37]. On the other hand, it has been shown that awareness of transfer techniques along with physical fitness training may reduce disability due to low back pain [4,38]. The majority of the participants (90%) in our study agreed that knowledge and awareness about the handling techniques improved after taking the course and were being applied in their daily tasks. However, the measurement is only subjective and an objective method such as video recording the handling techniques would be more reliable.

Although the content and format of the educational intervention delivered in the ergonomics workshop utilized the best available guidelines for patient handling and back care among nurses, ensuring the implementation of the acquired knowledge and skills into clinical practice was not feasible. This is mainly due to the inability to impact policymakers within the different organizations to supervise and ensure adherence to the workshop recommendations of safe practice. The risk assessment and control model that the workshop adopted requires commitment at all levels of the organization. This commitment needs to be visible where staff need to be involved in decisions. Thus, it is suggested that training on its own is not enough for bringing about change and must be supported with effective health and safety systems to ensure compliance with safe practice.

This study showed a connection between gained knowledge through an educational intervention received from an ergonomic workshop and decreased prevalence of occupational back pain among rehabilitation nurses in Saudi Arabia. Therefore, conducting the ergonomics workshop (educational intervention program) dedicated to increasing the level of knowledge about risk assessment and control of OBP among nurses should be encouraged. Thus, the results of this study can be generalized among rehabilitation nurses to prevent/avoid work-related physical injuries while conducting patient handling/care activities such as shifting, lifting, mobility, toilet care, transfer activities, and so on. In addition, the report of this study can be generalized among hospital/organizational policies makers, so that, the organizational policies might be more focused on clear, constructive directives “including international standardized guidelines” and vigilance in injury prevention for rehabilitation nurses involved in patients handling activities. Moreover, there is a need for “preventive plans including ergonomic advice” and a mechanism of their implementation in patient care units for avoiding occupational back pain and injuries among rehabilitation nurses.

### Limitation

Besides the value and importance of this study, there are few limitations also which require to be addressed in future studies. The participants tested their acquired knowledge by solving a questionnaire of 10 items of 10 marks provided to them. Their composite mean value was above average (7.67 ± 1.10 or 76.7%) which indicated a gain of knowledge because of attending the ergonomics workshop. Although, before filling the questionnaire, a written instruction was given to them to choose only the appropriate responses without using the workshop course manual. Similarly, a post-workshop evaluation at 6 months for the perceived knowledge was also conducted under strict supervision rather than believing in the honesty of the participants. However, this study did not try to ensure conducting the evaluation procedure at immediate and 6-months post-workshop under strict supervision using a professional standby video recording camera or recording through a CCTV or mobile camera. In addition, the study was limited to not including more nurses from more cities/provinces of the country. Therefore, a future study is required to address the shortcomings of this study to become part of the perfect solution for following safe patient handling and controlling the incidences of OBP among rehabilitation nurses. In addition, there is a need of conducting future studies focusing on the role of education and training in combating the barriers of safe patient handling and practices including working with overweight patients, more numbers of patients, being rushed or short-staffed, exhaustible working hours, certain physical requests from the patients that compromise nurse’s safety, and so on, among rehabilitation nurses.

## 5. Conclusions

The report of the study concludes by answering both questions/hypotheses. First, the level of knowledge about the risk assessments and control of OBP among rehabilitation nurses in Saudi Arabia increased after attending the ergonomic workshop focusing on work-related safety, risk assessments and control, and patient handling guidelines. Second, the increased level of knowledge after attending the ergonomics workshop since 6-months before markedly reduced the prevalence of OBP among the nurses. Importantly, the nurses learned and geared up themselves to practice the safe patient handling guidelines to avoid occupational back pain in the future. Additionally, the implication may further reduce the prevalence of disability (due to back pain), morbidity, and incurred costs, resulting in an overall improvement in activities of daily living (ADLs). Therefore, the rehabilitation nurses should update their knowledge and awareness about safety, risk assessments and control, and patient handling guidelines at regular intervals for increasing the knowledge and reducing the prevalence of OBP among them.

## Figures and Tables

**Figure 1 ijerph-18-08711-f001:**
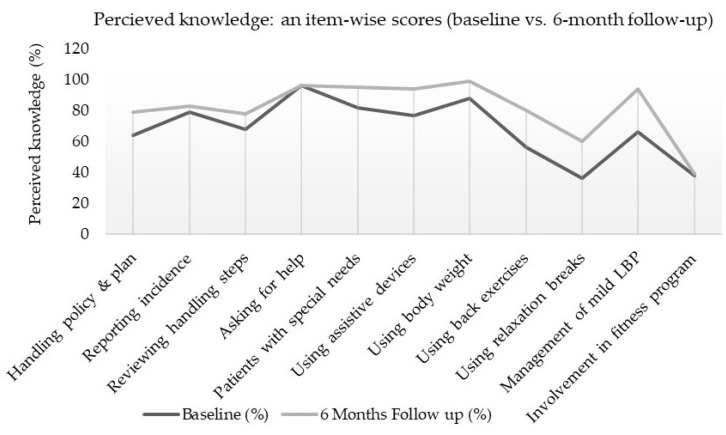
Item-wise comparison of perceived knowledge scores between baseline and 6-month follow-up.

**Figure 2 ijerph-18-08711-f002:**
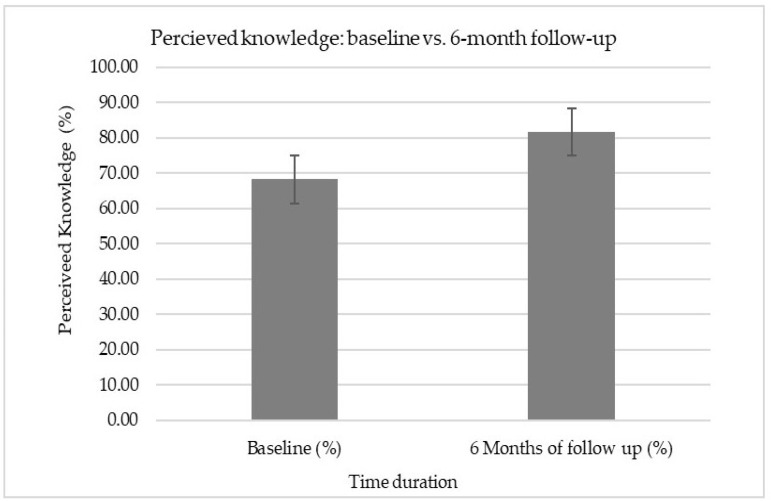
Overall comparison of perceived knowledge scores between baseline and 6-month follow-up.

**Table 1 ijerph-18-08711-t001:** Professional characteristic of the respondents.

Professional Characteristics	Respondents (%)
Educational level	
Diploma	25
Bachelor	52
Post graduate diploma	8
Clinical specialty	5
Other	1
Area of practice	
Medical units	1
Surgical units	31
Neurology/neurosurgery	2
Out-patient clinics	6
Intensive care units	3
Neuro rehabilitation	25
Other units (OBG, emergency etc.)	29
Rotations	3
Professional experience (years)	
2–4	7
4–6	7
6–8	13
8–10	16
>10 years	57

**Table 2 ijerph-18-08711-t002:** Item-wise differences in baseline vs. follow-up data of perceived knowledge.

Items of Perceived Knowledge (11-Items)	Baseline (%)	Follow Up (%) (6-Months)	Mean Difference (%) (95% CI)
Handling policy and plan	64	79	15
Reporting incidence	79	83	4
Reviewing handling steps	68	78	10
Asking for help	96	96	0
Patients with special needs	82	95	13
Using assistive devices	77	94	17
Using body weight	88	99	11
Using back exercises	56	80	24
Using relaxation breaks	36	60	24
Management of mild LBP	66	94	28
Involvement in fitness program	38	39	1
Overall total scores (mean ± SD) of perceived knowledge (%)	68.2 ± 19.2	81.6 ± 18.2	13.4 ± 9.5

## Data Availability

The data presented in this study are available on request from the corresponding author. The data are not publicly available due to reason of privacy.

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
