# Peer review of "Influence of Work-Related Safety and Health Guidelines on Knowledge and Prevalence of Occupational Back Pain among Rehabilitation Nurses in Saudi Arabia: A 6-Month Follow-Up Study"

_ijerph, 2021, doi:10.3390/ijerph18168711_

Round 1
Reviewer 1 Report
Introduction
page 2, 2º paragraph. "The literature is abundant with studies that used single factor intervention that showed limited impact on outcomes [8,9]." Please, cite more references.
page 2, 2º paragraph. Is know that the risk control process requires taking
all available steps to eliminate hazards. Please cite references about the three steps.
We agree that no enough studies that have addressed multidirectional strategies, including organized courses with practical workshops to improve knowledge and awareness about risk Assessments and their monitoring to reduce the prevalence of work-related low back pain among nurses, and that this study fills that shortage.
Methods
2.5. Procedure and measurements. Please provide the meaning of the abbreviation "CME"
Section Ethical Consideration. Please, detail these aspects: code and registration number, date, entity, etc.
Results
Figure 1. Please provide other higher quality figure.
Discussion
2º paragraph. "The literature is abundant with evidence on back injuries among nurses." Please cite refernces.
Please facilitate a structured discussion. Specifically include aspects of study limitations.
References
All references must include the doi and follow the journal's editing rules.
Reviewer 2 Report
Thank you for the opportunity to review this manuscript. The study of occupational back pain (OBP) within nursing is important, and it is clear that efforts should be made to help rehabilitation nurses reduce their back pain. With some revisions I believe that this manuscript would be publishable.
I encourage you to revise your Research Questions for clarity. The wording/grammar can be improved. Additionally, the Research Questions can better outline your variables (which are detailed later in the manuscript).
You mention mixed results of previous efforts to reduce occupational back pain. This study fits in with (and contributes to) existing research demonstrating a positive relationship between interventions and outcomes. That in itself is fantastic. However, I encourage you to document additional contributions of this study. That is, what else makes this study important and noteworthy? This can help further distinguish your research results.
As I read your manuscript I found myself wondering what barriers to engaging in safe practices nurses face. For example, this might include having lots of patients, being rushed or short staffed, working with patients who are overweight, having patients make certain physical requests that compromise nurse safety, and so on. Education and training certainly plays an important role (as your study demonstrates), but you might consider mentioning the study of barriers in as a topic for future research.
The manuscript is overall well organized and clear. However, the writing and grammar can use some work throughout to help provide clarity. For example, on page 2 you note, “In our knowledge, there is not enough studies that have approached multidirectional strategies, including organized courses with hands-on-workshop to improve the knowledge and awareness about risk assessments and its control in reducing the prevalence of work-related low back pain among nurses.” This could be revised to, “In our knowledge, there ARE not enough studies…” This is just one of a few areas that could be worked on, but these changes are certainly feasible.
The results of your study are important and I enjoyed reviewing your manuscript. I wish you the best of luck as you advance this toward publication.
Round 2
Reviewer 1 Report
I accept the manuscript.
Reviewer 2 Report
You have dutifully incorporated the recommended changes. I recommend some light attention to the use of English and overall grammar, but that can be handled in the copyediting process. Overall this manuscript is much improved and I commend you on your hard work.